# An independently tunable dual control system for RNAi complementation in *Trypanosoma brucei*

Raveen Armstrong[1], Matt J. Romprey[1], Henry M. Raughley[2], Stephanie B. Delzell[1¤a], Matthew P. Frost[1¤b], James Chambers[3], Grace G. Garman[3¤c], David Anaguano[1¤d], Michele M. Klingbeil[1,3]*

1 Department of Microbiology, University of Massachusetts, Amherst, Massachusetts United States of America, 2 Molecular and Cellular Biology Graduate Program, University of Massachusetts, Amherst, Massachusetts, United States of America, 3 Institute for Applied Life Sciences, University of Massachusetts, Amherst, Massachusetts, United States of America

¤a Current address: Latde Diagnostics, Life Science Laboratories, Amherst, Massachusetts, United States of America
¤b Current address: Department of Neuroscience, University of Connecticut Health, Farmington, Connecticut, United States of America
¤c Current address: Rensselaer Polytechnic Institute, Biomedical Engineering Department, Troy, New York, United States of America
¤d Current address: Cell Biology of Host-Pathogen Interaction Laboratory, Gulbenkian Institute for Molecular Medicine, Oeiras, Portugal
* mklingb@umass.edu

## Abstract

*Trypanosoma brucei* is a tractable protist parasite for which many genetic tools have been developed to study novel biology. A striking feature of *T. brucei* is the catenated mitochondrial DNA network called the kinetoplast DNA (kDNA) that is essential for parasite survival and life cycle completion. Maintenance of kDNA requires three independently essential paralogs that have homology to bacterial DNA polymerase I (POLIB, POLIC and POLID). We previously demonstrated that POLIB has a divergent domain architecture that displayed enzymatic properties atypical for replicative DNA polymerases. To evaluate the functional domains required for kDNA replication *in vivo*, we pursued an RNAi complementation approach based on the widely used tetracycline (Tet) single inducer system. Tet induction of RNAi and complementation with wildtype POLIB (POLIBWT) resulted in a 93% knockdown of endogenous *POLIB* mRNA but insufficient ectopic POLIBWT expression. This incomplete rescue emphasized the need for a more versatile induction system that will allow independent, tunable, and temporal regulation of gene expression. Hence, we adapted a dual control vanillic acid (Van)-Tet system that can independently control gene expression for robust RNAi complementation. Dual induction with Van and Tet (RNAi + Overexpression) resulted in 91% endogenous *POLIB* knockdown accompanied by robust and sustained ectopic expression of POLIBWT, and a near complete rescue of the *POLIB* RNAi defects. To more precisely quantify changes in kDNA size during RNAi, we also developed a semi-automated 3D image analysis tool to measure kDNA volume.

**Data availability statement:** All relevant data are within the manuscript and its Supporting Information files.

**Funding:** This work was financially supported by Bridge Funding from the University of Massachusetts College of Natural Sciences to M.M.K., a UMass Amherst Graduate School Pre-Dissertation Research Grant awarded to RA and the Donald P. Reed Legacy Fund. The funders had no role in study design, data collection and analysis, decision to publish, or preparation of the manuscript. The authors declare no competing financial interest.

**Competing interests:** The authors have declared that no competing interests exist.

Here we provide proof of principle for a dual inducer system that allows more flexible control of gene expression to perform RNAi and overexpression independently or concurrently within a single cell line. This system overcomes limitations of the single inducer system and can be valuable for elegant mechanistic studies in the field.

## Introduction

Kinetoplastids are early diverging protists of the Euglenozoa lineage that include free-living and parasitic examples. The pathogenic Trypanosomatids infect diverse species of insects, animals, and plants, but are best known for their medical and veterinary importance [1]. The African trypanosome, *Trypanosoma brucei* causes the neglected tropical disease human African trypanosomiasis and a related wasting disease in cattle called Nagana [2]. In addition to medical relevance, *T. brucei* has served as the model organism to address fundamental questions in eukaryotic biology due to its striking tractability with many advanced genetic tools to study molecular mechanisms [3,4].

Research on *T. brucei* has relied heavily upon the tetracycline (Tet)-On single inducer system first developed by Wirtz and Clayton that is based on stable transgenic cell lines expressing T7 RNA polymerase and Tet repressor [5]. Subsequent iterations of these plasmids led to a rapid expansion of molecular tools to include ectopic protein expression, conditional knockout strategies, and down-regulation of genes via RNA interference (RNAi) to quickly assess gene function [6,7]. RNAi remains a powerful tool in the field, used in both forward and reverse genetics including high-throughput genome wide screens, and has been instrumental in uncovering mechanisms associated with divergent trypanosome features [8,9].

One trypanosomatid property without counterpart in nature is their catenated mitochondrial DNA (mtDNA) network called kinetoplast DNA (kDNA) composed of two types of DNA molecules, maxicircles and minicircles within a single nucleoid [10,11]. Unlike the asynchronous replication of multiple mtDNA nucleoids that takes place in mammalian cells, replication of all the maxicircles and minicircles is coordinated to occur once each cell cycle. Synthesis of kDNA initiates prior to nuclear S phase shortly after flagellar basal body duplication and network replication completes before mitosis [12]. One distinctive feature of the kDNA replication mechanism is the topoisomerase II-mediated release of minicircle monomers from the network to replicate as free molecules. Newly replicated minicircles still containing nicks and gaps are then reattached to the network, while maxicircles replicate still catenated within the network [13]. During the later stages of network replication, the remaining nicks and gaps are repaired, the network undergoes scission, and progeny networks are then segregated into daughter cells via physical links with the basal body [14]. Another distinctive feature is the final physical connection between daughter kDNA networks at the later stages of segregation called the nabelschnur (umbilical cord). This filamentous bridge contains maxicircle threads [15] with at least two essential proteins for kDNA segregation, a leucine aminopeptidase (LAP1) and NAB70 [16,17].

The complexity of kDNA is reflected not only in the unique structure and replication mechanism, but also in the many additional proteins required for its maintenance. While most eukaryotes rely on one essential mtDNA polymerase (Pol γ), *T. brucei* uses three family A DNA polymerase paralogs (POLIB, POLIC, POLID) that are independently essential to maintain the kDNA network [18–22]. However, which of the paralogs is a processive, proofreading Pol, and how each contributes to the division of labor at the replication fork remain unclear.

Similar to other proofreading replicative Pols, *T. brucei* POLIB and POLID have predicted exonuclease (Exo) and Pol domains. POLIB is unique among all family A DNA Pols with the AlphaFold model revealing that the Exo domain is embedded within the Pol domain as an insertion in the thumb subdomain [23]. Additionally, recombinant POLIB displayed robust Exo activity (degradation) that prevailed over Pol activity (extension) on DNA substrates, with increased extension activity on RNA substrates [23]. These properties are atypical for a replicative enzyme using an Exo domain for proofreading. We hypothesize that the divergent features of POLIB facilitate a specialized role in kDNA maintenance.

An established approach for studying the functional domains within proteins is RNAi complementation in the single inducer Tet-On system [24–30]. Using this approach, we identified essential roles for two independent domains of POLIC; nucleotidyl incorporation for the Pol domain and a non-catalytic role in kDNA distribution for the N-terminal uncharacterized region (UCR) [22]. This structure-function analysis also revealed that the UCR was essential for proper localization during kDNA S phase. There are anecdotal reports that the single inducer system has failed for studies of other proteins, however these negative results are rarely published (personal communications). To expand the genetic toolbox and increase experimental flexibility, alternate inducible systems have been developed. A cumate inducible system was used in combination with the Tet-On system for inducible gene expression of ectopic proteins [31]. An isopropyl β-D-1 thiogalactopyranoside system was developed for inducible expression of tRNA [32], and a glucosamine-responsive riboswitch system has been reported for inducible gene silencing [33]. Only the vanillic acid (Van) system was reported for inducible expression of ectopic protein and dsRNA for RNAi mediated knockdown [34]. However, these systems have never been fully tested as a robust dual and independently inducible approach for RNAi complementation.

Here we demonstrate that the single inducer Tet-On system is not suitable for RNAi complementation of the kDNA polymerase POLIB, and establish the alternative Van-Tet dual inducible system [34] for procyclic form *T. brucei* as proof of principle for functional complementation of *POLIB* RNAi. Van-induced *POLIB* RNAi caused progressive kDNA loss that was detected earlier than previously published results. To track the earliest decline in kDNA size we developed a quantitative tool for the volumetric analysis of kDNA to aid in studying kDNA replication defects. The dual induction system combined with a more quantitative microscopy approach enables the study of challenging phenotypes like those that might be associated with partial kDNA replication defects encountered during structure-function analyses. These tools overcome key limitations and opens new avenues in trypanosomatid functional genomics.

## Materials and methods

For primer sequences refer to S1 Table.

### DNA constructs

**Vanillic acid inducible *POLIB* RNAi (pSLIB^Van).** To facilitate cloning the POLIB inverted repeats from pSLIB [20], the BglII site of pJ1271 [34] was modified to an MluI site using mutagenic primers (S1 Table) and the QuikChange Lightning Site Directed Mutagenesis Kit (Agilent) according to manufacturer's protocol to create pJ1271V2. The POLIB stemloop insert was excised from pSLIB using HindIII and MluI and ligated into pJ1271V2 to create pSLIB^Van.

**Tetracycline inducible expression of POLIB recoded variant.** Generation of the RNAi resistant POLIB-PTP tagged inducible expression construct pLew100-FLPOLIB_Rc-PTP^Puro was previously described [23]. The full-length POLIB open reading frame including the recoded region (249−801 bp) along with the PTP tag was PCR amplified from

pLew100-FLPOLIB$_{Rc}$-PTP$^{Puro}$ using primers listed in S1 Table for subsequent Gibson assembly (NEB) with pLEW100v5 to create pIBWTrecPTP$^{Phleo}$.

**Allelic tagging and single allele deletion of *POLIB*.** The *POLIB* C-terminal coding sequence (2261 bp) was PCR-amplified from *T. brucei* 427 genomic DNA using ApaI and EagI containing primers (S1 Table). The PCR fragment was ligated into ApaI and NotI sites of pC-PTP-NEO [35] to create pPOLIB-PTP$^{Neo}$. A 518 bp *POLIB* 5' UTR fragment was PCR-amplified using XhoI and HindIII containing primers and ligated into pKO$^{Puro}$ [36] creating pKOIB5'UTR$^{Puro}$. Subsequently, a 314 bp *POLIB* 3' UTR fragment was PCR-amplified using SpeI and XbaI containing primers and ligated into the downstream polylinker portion of the pKOIB5'UTR$^{Puro}$ to create pKOPOLIB$^{Puro}$.

## Trypanosome cell culture and transfection

*Trypanosoma brucei brucei* Lister 427 procyclic cells were cultured at 27°C in SDM-79 medium supplemented with 15% heat-inactivated fetal bovine serum. Strain 29-13 that harbors T7 RNA polymerase and tetracycline (Tet) repressor was cultured at 27°C in the same medium supplemented with G418 (15 µg/ml) and hygromycin (50 µg/ml). Transfected cell lines were additionally supplemented with the appropriate selectable drug. All transfected cell lines used in this study were clones obtained by limiting dilution that were selected based on protein expression levels and/or *POLIB* knockdown unless specified otherwise. For cell lines used in this study and associated modifications, refer to S2 Table.

## Tetracycline single inducer system

Generation and characterization of the parental POLIB RNAi cell line SLIB 2C7 was previously described [20]. Generation of the wildtype POLIB-PTP inducible overexpression cell line was previously described [23]. Clonal cell line P1D12 was chosen for this study and named IBOE$^{Tet}$. To generate the POLIB complementation cell line, SLIB 2C7 was transfected into the rRNA spacer region with NotI linearized pLew100-FLPOLIB$_{Rc}$-PTP$^{Puro}$ by nucleofection using the Amaxa Nucleofection Parasite Kit (Lonza) and selected with 1 µg/ml puromycin (Puro). Clonal cell line P2G7 was chosen for this study and named IBComp$^{Tet}$.

## Dual inducer system

The SMUMA (**S**ingle **M**arker **UMA**ss) cell line carrying T7 RNA polymerase, Tet repressor and Van repressor was generated by transfection of pJ1173 (gift from Jack Sunter) [34], into the tubulin locus of Lister 427 procyclic cells and selected with 1 µg/ml Puro. A single clone (P2B7) was selected and used for all subsequent transfections (S2 Fig). The Van-inducible POLIB RNAi cell line (IBRNAi$^{Van}$) was generated by transfecting pSLIB$^{Van}$ into SMUMA cells and selecting the cells with 10 µg/ml blasticidin. P2G9 clone was selected for subsequent transfections (IBRNAi$^{Van}$). To chromosomally tag POLIB, IBRNA-i$^{Van}$ was transfected with pPOLIB-PTP$^{Neo}$ and selected with 50 µg/ml G418. Cell population of IBRNAi$^{Van}$:IBPTP was used to evaluate the amount of protein remaining after RNAi and for the washout experiment described below. To complete the dual inducer system, IBRNAi$^{Van}$ was separately transfected with pIBWTrecPTP$^{Phleo}$, and stable transfectants were selected with 2.5 µg/ml phleomycin. Following dilution cloning, clone P2D11 and P5D1 were chosen for further characterization.

## Single expressor POLIB-PTP cell line

*T. brucei* 427 cells were transfected with the pKOPOLIB$^{Puro}$ XhoI/XbaI fragment (3047 bp). Cells were selected with 1 µg/ml Puro. Proper chromosomal integration was verified by Southern blot analysis. Clonal cell line P1C2 was then transfected with AatII linearized pPOLIB-PTP$^{Neo}$ and selected with 50 µg/ml G418. Single expressor clonal cell line P2F11 was used as a loading control for endogenous levels of POLIB.

## Inducible expression

In the single inducer system, RNAi and/or expression of protein were induced by addition of Tet (1, 2 or 4 µg/ml). In the dual inducer system, RNAi was induced by addition of Van (250 µM dissolved in DMSO) and expression of protein was

induced by addition of Tet (4 µg/ml). In both systems, cultures were supplemented daily with Van and/or Tet to maintain RNAi expression and POLIB-PTP tagged protein expression [24,34]. To evaluate whether Van-induction could be reversible, IBRNAi$^{Van}$:IBPTP cells were grown for 4 days in the presence of 250 µM Van, pelleted, and then washed once in PBS to remove the inducer. Cells were resuspended and grown in media lacking Van for an additional 2 days. Cells were then harvested at indicated time points for SDS-PAGE and western blot analyses.

## RNA isolation and northern analysis

Total RNA was isolated from 5 x 10$^7$ cells with TRIreagent (Sigma-Aldrich). RNA was fractionated on a 1.5% agarose/7% formaldehyde gel and transferred to GeneScreen (Perkin Elmer) as previously described [19]. Specific $^{32}$P-labeled probes were generated using 50 ng of PCR products for POLIB, POLIC and POLID (see S1 Table for primers) using the Random Primers DNA labeling system (Invitrogen) according to manufacturer's instructions. Standard hybridization and washing conditions were as previously described [37]. Specific mRNAs were detected using a Typhoon FLA 9500 Phosphorimager (GE Healthcare), and signals were quantified using ImageJ (http://imagej.nih.gov/ij/) and normalized against the ribosomal RNAs or tubulin signal.

## DNA isolation and Southern blot analysis

Total DNA was isolated from 1 x 10$^8$ cells using the Puregene Core Kit A (Qiagen) and free minicircles were analyzed as previously described [19,20]. Quantification was performed using a Typhoon FLA 9500 Phosphorimager (GE Healthcare) with background subtraction and signal intensities normalized against genomic DNA signal using ImageQuant TL Toolbox v8.1 software.

## SDS-PAGE and western blot analysis

Cells were harvested and washed using 1X PBS supplemented with protease inhibitor cocktail (1X cOmplete™ EDTA-free protease inhibitor (Roche)). Samples were fractionated by SDS-PAGE and transferred overnight onto a PVDF membrane. Membranes were blocked in 5% non-fat dry milk in 1X Tris-buffered saline for at least 1 hr. PTP-tagged proteins were detected with peroxidase-anti-peroxidase soluble complex (PAP, 1:2000, Sigma). Tet repressor (TetR) was detected with TetR monoclonal antibody (clone 9G9, 1:1000, Takara). Loading control was detected with anti-EF1α1 for 1 hr (1:15000, Santa Cruz Biotechnologies). Incubation with HRP conjugated goat anti-mouse (1:2500, Sigma) for 1 hr was performed for both EF1α1 and TetR antibodies. SuperSignal West Pico Plus chemiluminescent substrate (ThermoFisher) was used for protein detection. Band intensities were quantified using ImageJ software.

## Immunofluorescence microscopy

Cells were harvested at 1000 x *g*, resuspended in 1X PBS, and adhered to poly-L-lysine (1:10) coated slides (5 min). Cells were fixed in 3% paraformaldehyde (5 min), washed twice in 1X PBS containing 0.1M glycine and then permeabilized for 5 min with 0.05% Triton X-100 or methanol overnight. Cells were then washed thrice with 1X PBS. PTP-tagged protein was detected with rabbit polyclonal anti-protein A (1:1000, 1 hr, Sigma) followed by secondary antibody Alexa Fluor 488 goat anti-rabbit (1:250, 1 hr, ThermoFisher). DNA was stained with 1 µg/ml of 4', 6-diamidino-2-phenylindole (DAPI) for 5 min, washed thrice in 1X PBS and mounted in Vectashield (Vector Laboratories).

## Image acquisition and semi-automated analysis

Images were acquired using an inverted Nikon Ti2-E wide-field fluorescence microscope with a Nikon Plan Apo $_\lambda$ 60x 1.45 numerical aperture objective lens. Z stacks with a step size of 0.2 µm were deconvoluted using the Landweber algorithm within NIS Elements, a closed-source analysis tool developed by Nikon Instruments. The General Analysis 3 module was used to create Kinetometric 1.0, a custom quantification algorithm for kDNA volumetric analyses as follows. Briefly,

nuclear DNA and kDNA were segmented on DAPI images using the 3D threshold function. kDNA was distinguished from nuclear DNA using the 3D volume function with a set threshold range of 0–4 arbitrary units. The kDNA were then colorized and assigned unique identification numbers for ease of tracking 3D volume. Tables for each image were exported separately, and cell cycle stage was manually annotated based on basal body staining. Items that were excluded from the analysis include: cells that did not appear intact by phase contrast, DAPI spots within the nuclear DNA similar to the size range of a kDNA were excluded (thresholding artifact), and kDNA that was overlapping with the nuclear DNA. Intact cells with no detectable kDNA were manually annotated as 0.0 µm$^3$ volume. Statistical significance of results was calculated by an unpaired two-tailed t test.

## Results

### Tetracycline single inducer system

We reported previously that *TbPOLIB* silencing using a stemloop dsRNA led to a loss of fitness (LOF), progressive loss of kDNA network, and disruption of free minicircle replication intermediates, indicating that POLIB has an essential role in kDNA replication [20,21]. Recent enzymatic characterization of recombinant POLIB strongly suggests that the DNA polymerase activity is less robust than that of the three other mitochondrial paralogs and that the exonuclease activity may play a dominant role in kDNA replication [23]. To examine which of the domains are necessary for kDNA replication, we pursued a genetic complementation approach wherein dsRNA expression to silence *POLIB* was coupled with inducible ectopic overexpression (OE) of a recoded RNAi-resistant version of *POLIB,* both of which were simultaneously induced with Tet.

The Tet-On single inducer complementation system requires three separate cell lines in order to conduct the necessary control experiments: OE only, RNAi only, and complementation (RNAi + OE). OE of recoded PTP-tagged wildtype POLIB (IBWT) in the absence of *POLIB* RNAi resulted in no significant change in fitness using the standard 1 µg/ml Tet for induction (Fig 1A). Throughout the induction, IBWT levels were consistently 11-fold above endogenous POLIB levels detected in a cell line engineered to express PTP-tagged POLIB from a single allele (C) while no protein was detected in uninduced samples (Fig 1B). Additionally, IBWT was expressed homogenously in the clonal cell population and localized throughout the single mitochondrion as well as near the kDNA that is detected using DAPI (Fig 1C). A higher concentration of Tet (4 µg/ml) also did not significantly impact fitness (S1A Fig) despite the 17-fold increase in protein levels (S1B Fig).

To determine if the defects caused by *POLIB* RNAi [20] could be rescued by ectopic expression of IBWT, the parental cell line IBRNAi$^{Tet}$ was transfected with pLew100-FLPOLIB$_{Rc}$-IBPTP$^{Puro}$ to create the complementation cell line IBComp$^{Tet}$. Pilot experiments indicated that 1 and 2 µg/ml of Tet were insufficient to produce IBWT above endogenous POLIB levels (S1D Fig). Therefore, clonal cell line IBComp$^{Tet}$ P2G7 was grown in the presence of 4 µg/ml Tet to produce maximal expression of IBWT. Fitness of induced IBComp$^{Tet}$ was slightly improved over RNAi alone (IBRNAi$^{Tet}$) (13.3 vs 10.8 doublings; day 10). However, induced IBComp$^{Tet}$ still showed a reduced growth rate compared to its uninduced control (Un) (13.3 vs 17.3 doublings; day 10), indicating an incomplete rescue (Fig 1D). In contrast to OE alone, IBComp$^{Tet}$ cells revealed only a gradual increase in IBWT abundance over the induction period with a 2.4 fold increase above endogenous levels by day 10 (Fig 1E). Lastly, fluorescence microscopy revealed heterogenous and low expression of PTP-tagged POLIB even at day 10 during IBComp$^{Tet}$ induction. Additionally, aberrant kDNA network morphology was noted including small kDNA, no kDNA and asymmetrically divided kDNA (Fig 1F). Interestingly, 4 µg/ml Tet resulted in an increased mRNA knockdown compared to previously published data (93% vs 90%) and a faster LOF in RNAi alone after just 6.8 doublings (day 4) compared to 8.5 doublings for 1 µg/ml inductions (Fig 1D). The increased Tet concentration did not improve IBWT abundance nor fitness (S1C and S1D Fig). The incomplete rescue with IBWT was not suitable for a detailed POLIB structure-function study. We hypothesized that a more versatile system with independent, tunable, and temporal regulation of gene expression would overcome the limitations of the widely used Tet single inducer system.

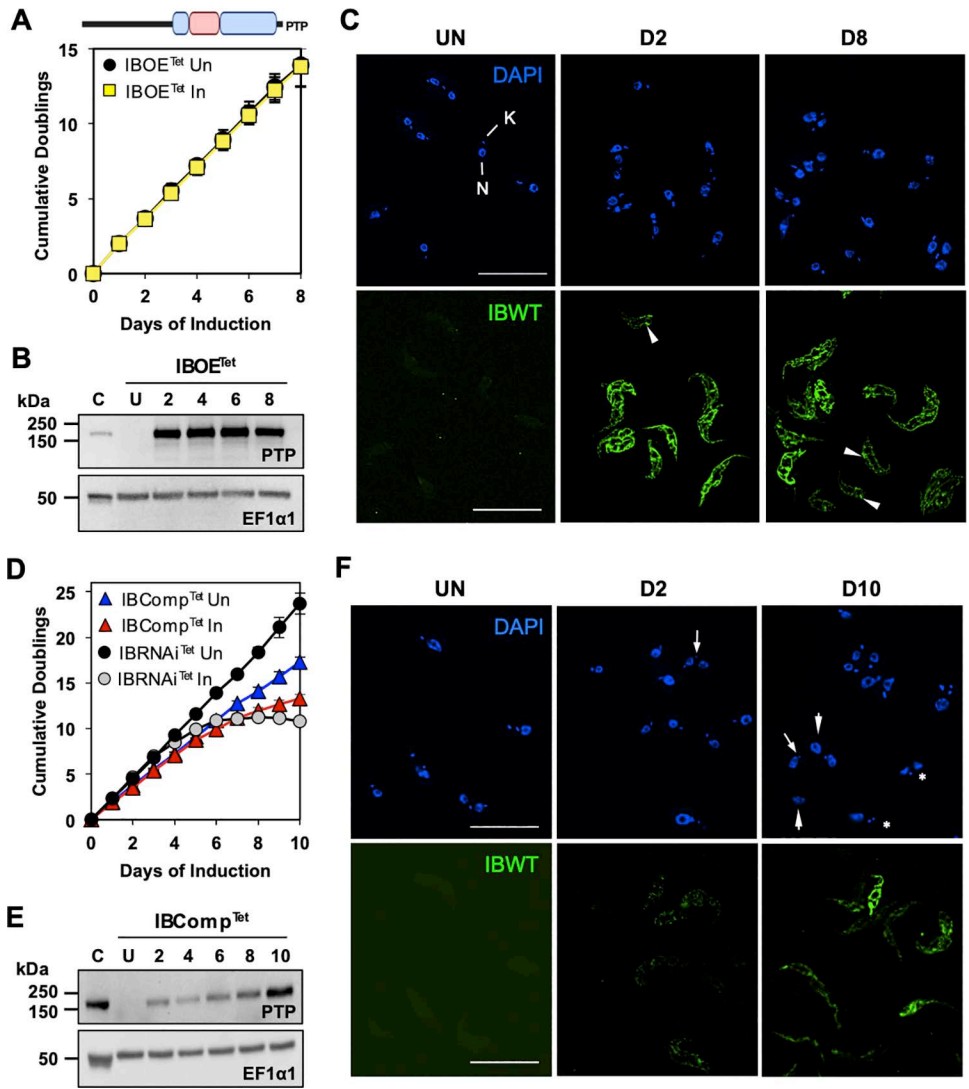

**Fig 1. Tetracycline single inducer system for POLIB RNAi complementation. (A)** Top; domain structure of TbPOLIB. Blue, Pol domain; pink, Exo domain. Bottom, growth curve of IBOE^Tet cell line grown in the absence and presence of 1 µg/ml Tet. **(B)** Western blot detection of IBWT and loading control EF1α1 during an 8 day induction of IBOE^Tet with Tet. C, POLIB-PTP single allele tagged cell line. $2 \times 10^6$ cell equivalents were loaded per lane. **(C)** Representative images of IBWT at selected IBOE^Tet induction points. DAPI staining (blue); anti-protein A (green). N, nucleus; K, kDNA. Arrowheads, POLIB signal accumulation near the kDNA. Brightness was increased for uninduced samples to highlight background fluorescence. **(D)** Growth curves of IBComp^Tet and IBRNAi^Tet cell lines grown in the absence and presence of 4 µg/ml Tet. **(E)** Western blot detection of IBWT and loading control EF1α1 during a 10 day induction of IBComp^Tet. C, POLIB-PTP single allele tagged cell line; $4 \times 10^6$ cell equivalents loaded; all other lanes, $2 \times 10^6$ cell equivalents. **(F)** Representative images of IBWT at selected IBComp^Tet induction points. DAPI staining (blue); anti-protein A (green). Arrowheads, cell with no kDNA; arrow, cells with small kDNA; asterisk, cells with unequal division of kDNA. Brightness was increased for uninduced samples to highlight background fluorescence. For **(A)** and **(D)**, error bars represent ±s.d. of the mean from four biological replicates. Some error bars are too small to be displayed. For **(C)** and **(F)**, size bar, 20 µm.

## Establishing a dual inducer system

To achieve independent control of gene expression required for POLIB RNAi complementation, we adapted the Van-inducible system to operate in parallel with the Tet based system. Lister 427 procyclic cells were transfected with pJ1173 [34] to create SMUMA (**S**ingle **M**arker **UMA**ss), a single marker cell line expressing T7 RNA polymerase, Tet

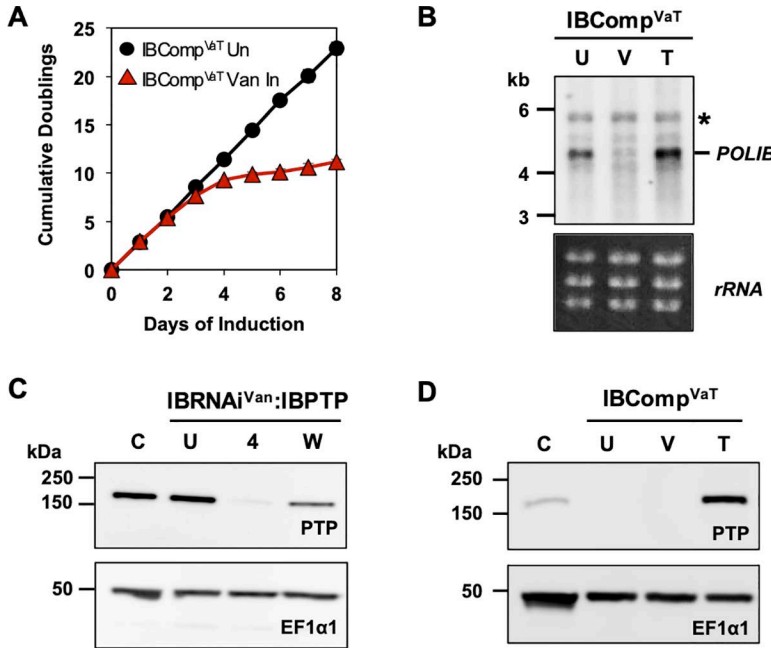

**Fig 2. Specific and independent vanillic acid induction of *POLIB* RNAi. (A)** IBComp^VaT cell line was grown in the absence or presence of 250 µM Van. Error bars represent ±s.d. of the mean from three biological replicates. Some error bars are too small to be displayed. **(B)** Northern blot of total RNA from IBComp^VaT. U, uninduced; V, induced with 250 µM Van for 48 hr; T, induced with 4 µg/ml Tet for 48 hr. Top, probing for *POLIB* mRNA; Bottom, EtBr-stained rRNAs as loading control. *, a non-specific cross-reacting band. **(C)** Western blot detection of IBPTP from IBRNAi^Van:IBPTP grown in 250 µM Van for 4 days (4). C, single allele cell line endogenously expressing POLIB-PTP; U, uninduced; W, inducer removed; 2 x 10^6 cell equivalents were loaded per lane. **(D)** Representative western blot detection of IBPTP and loading control EF1α1 from IBComp^VaT. 2 x 10^6 cell equivalents were loaded per lane. C, single allele cell line endogenously expressing POLIB-PTP. IBComp^VaT clonal cell line P2D11 was used.

repressor (TetR) and Van repressor (VanR). Western blot analyses revealed high levels of TetR protein from 5 clonal cell lines, with clone P2B7 expressing ~20 fold higher TetR protein than 29-13 cells (S2A Fig). Northern blot analyses revealed clone P2B7 also expressed highest levels of *VanR* mRNA (S2B Fig). We next tested whether maximal concentrations of Tet or Van impacted fitness of SMUMA cells. Addition of 4 µg/ml of Tet had no impact on the fitness of SMUMA cells. However, in the presence of 250 µM Van (in DMSO) there was a slight decline in fitness that started after 5 days of growth resulting in fewer doublings compared to the untreated cells (17.1 vs 18.1 doublings) (S2C Fig). Based on these results, clone P2B7 was selected as the parental SMUMA cell line.

A Van-inducible POLIB RNAi cell line (IBRNAi^Van) was generated by transfecting pSLIB^Van into SMUMA cells, and subsequently transfecting with pIBWTrecPTP^Phleo to create the dual inducer POLIB RNAi complementation system. The dual system uses the same stemloop *POLIB* dsRNA trigger and RNAi resistant IBWT inducible expression construct as in the single inducer system [20]. In addition to independent control over gene expression, other advantages include temporal control of gene expression, and all controls (RNAi only, OE only) can be performed in the same cell line decreasing the number of required cell lines and the inherent variability that accompanies separately transfected cell lines. Clonal cell line P2D11 was selected for further characterization based on 91% *POLIB* mRNA knockdown and ~10-fold increase in IBWT protein and will be referred to as IBComp^VaT (Fig 2B and 2D).

## Independent gene expression in IBComp^VaT cell line

To induce *POLIB* RNAi only, IBComp^VaT cells were treated with 250 µM Van and depletion was confirmed by northern and western blotting. The characteristic LOF was observed following 6–7 doublings that correlated with *POLIB* mRNA loss

and depletion of chromosomally PTP-tagged POLIB (Fig 2A and 2B). *POLIB* mRNA knockdown (91%) was evident 48 hr after Van induction of dsRNA synthesis with no significant change in mRNA levels when only Tet was added (Fig 2B). No significant changes in the two other essential mitochondrial paralogs (POLIC and POLID) were noted with the addition of Van (S3A Fig). Endogenous POLIB-PTP protein decreased by 97% after 4 days of Van induction with depletion starting as early as day 2 (Figs 2C and S3B). Removal of the Van inducer resulted in restoration of expression of chromosomally tagged POLIB-PTP reaching 54% compared to the uninduced control after growing the cells in Van-free medium for 48 hr (Fig 2C). IBComp^VaT cells were treated with 4 µg/ml Tet to induce expression of an ectopic copy of IBWT. OE in the absence of *POLIB* RNAi resulted in a 6-fold increase in IBWT over the control while Van only induction produced no IBWT protein (Fig 2D) confirming that the two inducers operate independently and do not interfere with each other's function.

## Volumetric analysis of kDNA networks

Previously, *POLIB* RNAi in the single inducer system resulted in a progressive shrinking and loss of kDNA that was determined subjectively [20]. Size was analyzed (normal, small and no kDNA), while volume was not. For more robust analyses of progressive kDNA loss and other changes during RNAi in IBComp^VaT cells, we developed a semi-automated analysis pipeline (Kinetometric 1.0) within NIS Elements that could quantify changes in kDNA volume with decreased experimentalist bias.

To validate this new approach, we analyzed 588 cells in an asynchronous population of the dual inducer parental cell line SMUMA. First, we determined the stages of the kDNA duplication cycle from deconvoluted images using well-established DAPI staining and basal body position as markers for cell cycle progression [38], and then applied Kinetometric 1.0. Cells exhibited normal duplication and segregation of nuclear and kDNA with each cell cycle karyotype being easily observed. 31.7% of the cells had a single unit kDNA (1N1K), 47.0% were undergoing kDNA replication (1N1K*), 12.7% had already replicated and segregated their kDNA (1N2K) and 8.6% had completed kDNA segregation and mitosis (2N2K cells) (S4 Fig). These results are comparable to previously published work [15,36,39,40]. The mean kDNA volume for G1 cells (1N1K) was 1.74 µm$^3$ while cells undergoing kDNA S phase (1N1K*, now containing 2 basal bodies) showed a ~ 1.7-fold increase in the mean kDNA volume (2.62 µm$^3$). Lastly, cells with two segregated progeny networks prior to and post mitosis (1N2K, 2N2K) have kDNA networks that correspond to mean kDNA volumes of 1.83 and 1.83 µm$^3$, respectively (S4 Fig). During kDNA S phase, the network doubles in size before segregation into two daughter networks and this is reflected in our volumetric analysis. Therefore, we applied volumetric analysis to assess the effects of *POLIB* silencing on kDNA networks.

A hallmark of a kDNA replication defect is an overall progressive decrease in kDNA network size [18–20,37,41–43]. Following 8 days of Van induction to characterize RNAi only, IBComp^VaT cells began to display reduced kDNA volume as early as day 2. Representative examples of DAPI-stained uninduced and Van induced IBComp^VaT cells with normal and progressive changes in kDNA during *POLIB* RNAi are shown (Fig 3A). Using Kinetometric 1.0, we analyzed kDNA volume independent of cell cycle stage from >300 SMUMA cells and >650 cells from selected induction points during *POLIB* RNAi. Uninduced IBComp^VaT cells displayed a mean kDNA volume of 2.00 µm$^3$ compared to 2.15 µm$^3$ for the SMUMA parental cell line (Fig 3B). Throughout the RNAi induction, there was a progressive decline in the mean kDNA volume that was evident after just 2 days of *POLIB* RNAi with a 19% decrease in mean kDNA volume detected (1.62 µm$^3$) that plateaued by day 4 (1.61 µm$^3$). By day 6, the mean kDNA volume decreased by 30.5% (1.39 µm$^3$) and continued to decline at day 8 (1.30 µm$^3$). Kinetics of kDNA loss indicates an accumulation of at least two subpopulations, one with 50% smaller kDNA (0.44-0.87 µm$^3$) and another with ultra small kDNA (75% smaller, 0.00-0.44 µm$^3$) compared to SMUMA 1N1K cells. While the decline in mean kDNA volume was statistically significant, there were other notable changes. There was an accumulation of 2N1K cells, 1N0K cells, and cells with unequal segregation of the kDNA (see representative images Fig 3C). There were also notable changes in the morphology of the nucleus. Many nuclei appeared elongated after day

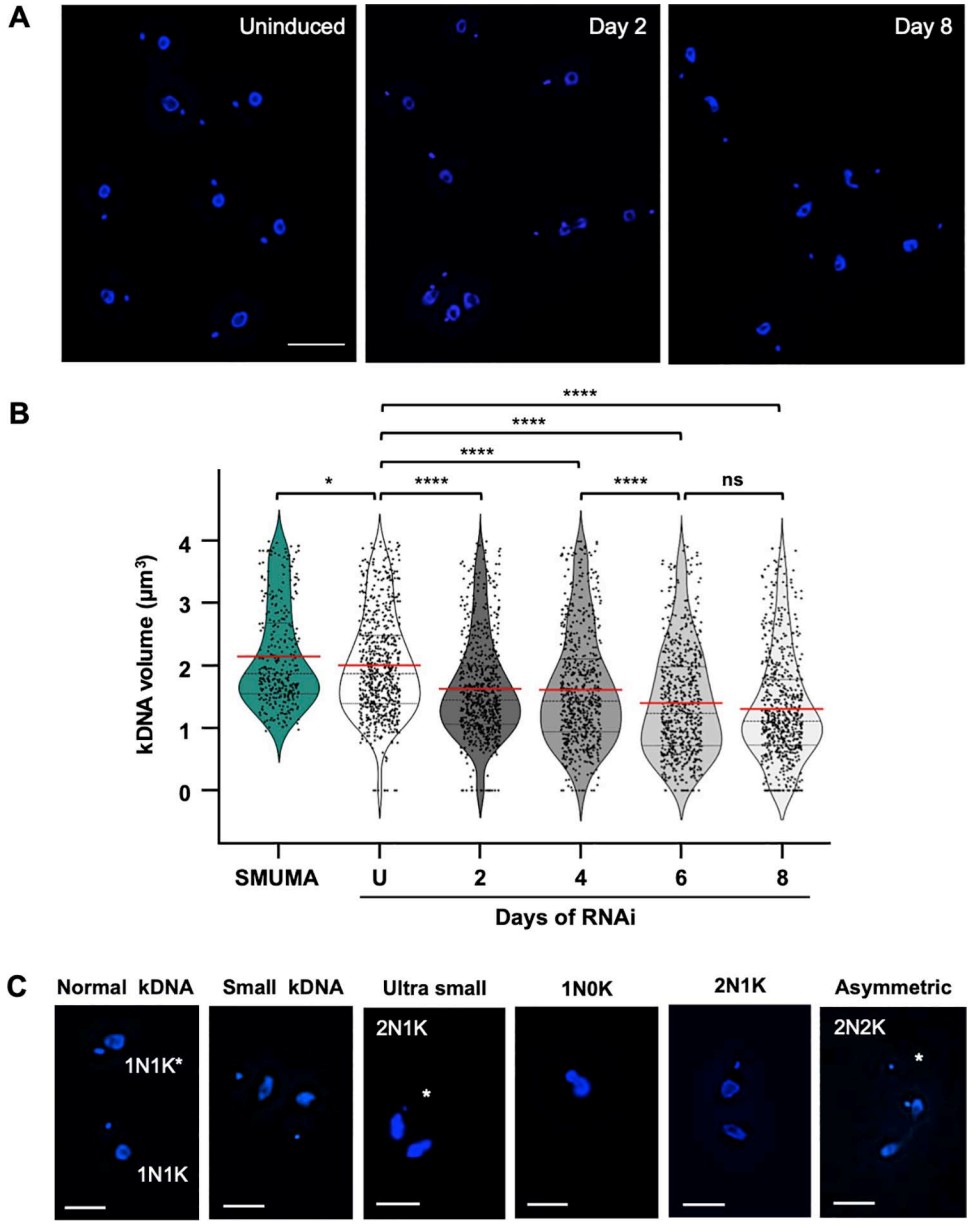

**Fig 3. Impact of *POLIB* RNAi on kDNA volume. (A)** Representative images at selected induction points of IBComp^Vat (P2D11 and P5D1 clones) induced with 250 µM Van. Scale bar, 20 µm. **(B)** Quantification of kDNA volume from DAPI-stained cells for each data point (>300 for SMUMA and >650 for IBComp^Vat). The red line indicates the mean of the kDNA volume. ns, not significant ($P$ value ≥0.05); *, $0.01 \leq P < 0.05$; ****, $P < 0.0001$, as calculated by an unpaired two-tailed $t$ test. **(C)** Representative images for various kDNA phenotypes. Scale bar, 10 µm.

4 of *POLIB* RNAi, some with increased DAPI staining and 2.4% of day 8 population were zoids (anucleated cells with kDNA). Both morphologies are common with cell cycle defects [44,45].

## Disruption of minicircle replication

As previously reported, *POLIB* silencing not only resulted in the progressive loss of the kDNA network but also disrupted the pattern of minicircle replication intermediates [20,21]. Although minicircle replication intermediates are a small fraction

of the total kDNA in an asynchronous population [46], free minicircles are resolved in a specific and predictable pattern on a single-dimension agarose gel containing ethidium bromide that can be quantified using Southern blotting with a minicircle-specific probe [20,41,42]. RNAi induced IBComp[VaT] cells resulted in a progressive increase in unreplicated covalently closed (CC) monomers in parallel with the accumulation of nicked/gapped progeny (N/G) and a linear species (S5A Fig) confirming a disruption in minicircle replication.

## Dual inducer RNAi complementation

One advantage of the dual inducer system is that all experiments (critical controls) can be performed in the same cell line as complementation. As one of these controls, overexpression of IBWT was evaluated. Ectopic expression of IBWT by the addition of 4 µg/ml Tet resulted in no significant change in fitness (Fig 4A). Similar to the single inducer system, IBWT levels were consistently above endogenous POLIB levels (6-fold increase), and IBWT was expressed homogenously in the clonal cell population and localized throughout the single mitochondrion as well as near the kDNA (Fig 4B and 4C). Moreover, no changes to the kDNA network and free minicircle replication intermediates were noted (S5C Fig). The dual inducer system allows for independent and/or simultaneous inductions. To achieve RNAi complementation, IBComp[VaT] cells were grown in the presence of both Van and Tet. Dual induction resulted in a near complete rescue of fitness with doublings reaching 20.9 after 8 days of dual induction compared to 23.0 for uninduced cells (Fig 4A). Similar to RNAi only, endogenous *POLIB* mRNA was depleted 86–99% throughout the 8 day induction (Fig 4D). Consistent and robust expression of IBWT was detected with a 7-fold increase that appeared homogeneous throughout the population (Fig 4E and 4F). Complementation with IBWT alleviated the defects in kDNA volume as well as cell cycle disruption. Analysis of free minicircles by Southern blotting showed minimal changes in the pattern of free minicircles with no accumulation of linear species. Together these results confirm the specificity of *POLIB* RNAi and establish proof of principle for functional RNAi complementation using a dual inducer system in procyclic form *T. brucei.*

## Discussion

In trypanosomes, the single mitochondrial nucleoid containing the kDNA replicates once per cell cycle in a highly coordinated process. The maintenance of this network relies on multiple proteins, many of which localize to subcompartments surrounding the kDNA network [47,48]. Additionally, paralogs with similar enzymatic activities but non-redundant roles in kDNA maintenance were identified. Among these are two topoisomerases, two origin binding proteins, two primases, two ligases, six Pif1 helicases and six DNA polymerases all of which are exclusively localized to the mitochondrion [18,37,41,42,49–52]. Divergent features of these paralogs likely facilitate specialized roles in kDNA maintenance.

RNAi is a powerful functional genomics tool to study trypanosome biology that has been widely used to understand the replication and segregation of kDNA [10,53]. While 5 classes of RNAi mutants for kDNA replication and segregation have been identified to facilitate a deeper understanding of these unusual processes [54], there remains the challenge of assigning roles of proteins with subtle or pleiotropic phenotypes. Previously we had success using the Tet-ON single inducer RNAi complementation system to demonstrate that individual domains of the kDNA polymerase POLIC contributed to separate roles in kDNA maintenance (nucleotidyl incorporation and progeny distribution) clarifying the original pleiotropic RNAi phenotype [22].

POLIB is unique among all family A DNA Pols with an Exo domain inserted within the Pol domain. POLIB also exhibited Exo activity that unexpectedly prevailed over Pol activity on DNA substrates with recombinant and immunoprecipitated POLIB [23]. To dissect the individual biological roles of the POLIB domains, we sought to use the established single inducer RNAi complementation system (IBComp[Tet]). During *POLIB* RNAi induced with 4 µg/ml Tet, kDNA loss began as early as day 2 and LOF started days 3–4 post induction of *POLIB* dsRNA. However, during complementation the levels of ectopic POLIB did not reach endogenous control levels until day 10, well after the depletion of POLIB had impacted the kDNA network. Unfortunately, the insufficient ectopic expression of wildtype POLIB could not rescue the RNAi phenotype

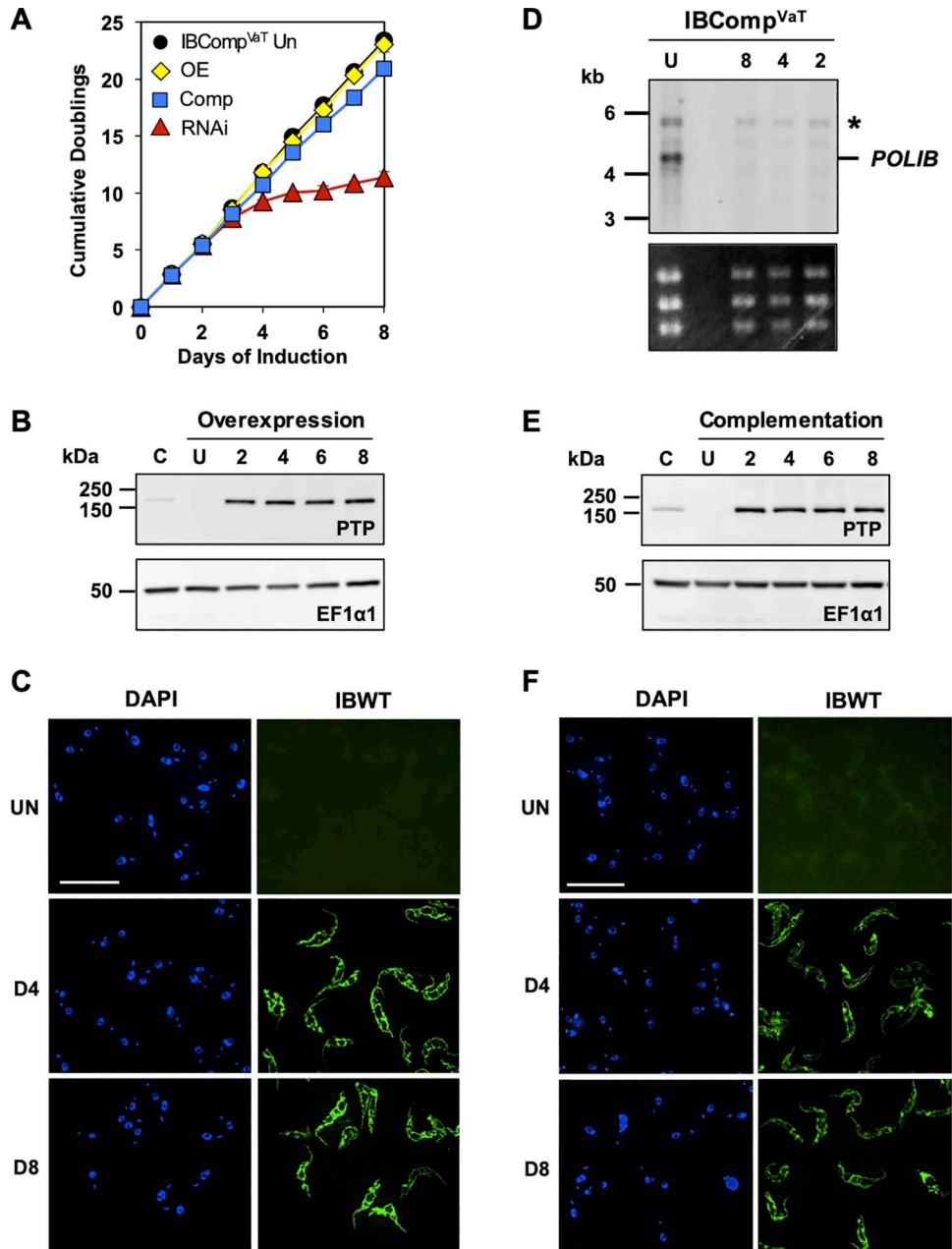

**Fig 4. Dual inducer system for *POLIB* RNAi complementation. (A)** IBComp^VaT was grown in the absence and/or presence of 250 μM Van and/or 4 μg/ml Tet. Error bars represent ±s.d. of the mean from three biological replicates. Some error bars are too small to be displayed. **(B)** Representative western blot detection of IBPTP and loading control EF1α1 from IBComp^VaT grown in 4 μg/ml Tet for 8 days. 2 x 10^6 cell equivalents were loaded per lane. C, single allele cell line endogenously expressing POLIB-PTP. **(C)** Representative images of IBWT for IBComp^VaT Tet-induced at selected induction points. DAPI staining (blue); anti-protein A (green). Brightness was increased for uninduced samples to highlight background fluorescence. **(D)** Northern blot of total RNA from IBComp^VaT. U, uninduced, or induced with 250 μM Van and 4 μg/ml Tet at selected induction points. Top, probing for *POLIB* mRNA; Bottom, EtBr-stained rRNAs as loading control. **(E)** Representative western blot detection of IBWT and loading control EF1α1 from IBComp^VaT grown in 250 μM Van and 4 μg/ml Tet for 8 days. C, single allele cell line endogenously expressing POLIB-PTP; 2 x 10^6 cell equivalents were loaded per lane. **(F)** Representative images of IBWT for IBComp^VaT Van and Tet-induced at selected induction points. DAPI staining (blue); anti-protein A (green). Brightness was increased for uninduced samples to highlight background fluorescence.

(Fig 1D–1F). The failure of the IBWT rescue demonstrated that the single inducer Tet-ON system was not suitable for study of POLIB, and may indicate that this system would not work well to study other abundant proteins. POLIC is the least abundant of the paralogs that could explain why *POLIC* RNAi complementation was successful. To further study POLIB or other more abundant proteins associated with essential processes, we focused on establishing a dual induction system that could provide independent, tunable, and temporal regulation of gene expression for detailed mechanistic studies.

We first generated SMUMA by engineering a Lister 427 procyclic cell line to express T7RNAP along with two regulatory elements, TetR and VanR, based on the original approach reported by Sunter [34]. This single marker cell line was then adapted for a dual control RNAi complementation system by transfection of plasmids for Van-inducible RNAi and Tet-inducible OE, allowing independent regulation of both (Fig 2). Simultaneous induction of RNAi (Van) and OE (Tet) resulted in 91% *POLIB* mRNA knockdown and ~10-fold increase in ectopic protein expression, levels that were nearly identical to those achieved during independent inductions (Fig 4D and 4E). Importantly, dual induction resulted in a near complete rescue of the RNAi defects (Figs 4A, 4F, and S5B). Fitness for *POLIB* RNAi complementation lagged slightly behind the uninduced control but was likely due to the solvent carrier of Van (DMSO) that impacted the growth rate of SMUMA cells (Figs 4A and S2C). This study provides proof of principle that the dual control RNAi complementation system is robust and can be used for structure-function studies of proteins that fail using the single inducer system. Similar dual control elements have been previously reported [31,34], but this study represents the first demonstration of successful use of one for RNAi complementation.

The dual induction system relies on the parental strain SMUMA that has a faster doubling time (10 hr) and expresses higher levels of TetR compared to the commonly used 29–13 strain for the single inducer system (13.5 hr) (S2A Fig). 29-13 cells are widely used and may have undergone additional modifications over time. Other studies also reported generation of faster growing transgenic 427 derivatives that also express higher levels of TetR than 29-13 cells [31,34,55,56]. The strain differences likely influence phenotypic outcomes despite similar experimental conditions. For instance, Van-induced *POLIB* RNAi results in a faster LOF (7.7 doublings) compared to the previously reported single inducer *POLIB* RNAi (9.3 doublings) (Fig 4A) [20]. Additionally, instead of a majority of the cells lacking kDNA after POLIB depletion in the Tet-ON system, Van-induced *POLIB* RNAi resulted in an accumulation of cells with small kDNA (Fig 3).

Historically, quantifying kDNA loss *in vivo* has been challenging due to limited tools. Visual assessment of kDNA size is subjective, time consuming and not comprehensive. Previous studies have performed quantitative analysis of kDNA size in 2D micrographs using ImageJ/Fiji [57–60]. While these provide some detail, they do not capture the three-dimensional organization of the kDNA network. Additionally, they fail to account for variations in kDNA size across the z-plane nor do they account for cell orientation during imaging. These factors can lead to misrepresentation of the true kDNA network size. To address these limitations, we developed Kinetometric 1.0, a quantitative tool to analyze cells in an unsynchronized population and determine kDNA volume irrespective of cell cycle stage. The kDNA is a structurally complex network that undergoes continuous topological remodeling to accommodate replication, repair, and segregation. Analyses using Kinetometric 1.0 validated the inherent volumetric changes that occur during the kDNA replication cycle, establishing a quantitative framework to study kDNA-associated defects (S4 Fig). This new pipeline minimizes time for analyses and provides a biologically significant metric to capture subtle yet significant changes in kDNA size that 2D projections cannot offer.

Previously, the progressive loss of kDNA was not detected until the cells started to exhibit LOF [20]. In the current study, the *POLIB* RNAi kDNA phenotype was detected as early as day 2 (prior to LOF) with a significant decrease in mean kDNA volume that included the accumulation of 2 subpopulations (small and ultra small) accounting for 10.8% of the total population. While there was no significant change in mean kDNA volume between day 2 and day 4, there was a notable shift in the 2 subpopulations that represented 21% of the total cells at the time when LOF was first noted. Although not reported in this study, the pipeline also captures other quantitative data, including inter-organellar distance. Volumetric

analysis of kDNA can serve as a sensitive and robust reporter of kDNA replication defects, and the framework developed for Kinetometric 1.0 could be applied to other trypanosomatid species, broadening its impact on the field.

One consideration for the current version of our pipeline is that it does not account for cells where there is overlap of the kDNA and nuclei DAPI signals that prevent accurate segmentation. Differential staining of the nuclei and kDNA with propidium iodide and DAPI, respectively, combined with color deconvolution [58] may address this limitation by providing clear segmentation to precisely measure the volumes of the two genomes. This will also improve thresholding accuracy by reducing artifacts, such as the DAPI spots within the nuclear DNA that resemble kDNA in size. To streamline analyses, we are also expanding the pipeline to automatically assign cell cycle status using cell membrane markers and to track newly synthesized kDNA through *in situ* TdT labeling. Kinetometric 1.0 in combination with the dual inducer system provides a powerful approach for robust structure-function analyses to evaluate POLIB's *in vivo* enzymatic roles.

While the single inducer Tet-On system has been a valuable tool in advancing functional genomics in *T. brucei*, it appears to be insufficient for RNAi complementation studies of abundant proteins such as POLIB (this study) or POLID (unpublished results). Our study provides proof of principle for a robust dual control RNAi complementation system that leverages both the Tet and Van-inducible expression systems. The dual control system provides major advantages to the field. It allows for independent and tunable inducible gene expression to control multiple processes. The dual inducer system also enables temporal regulation to modulate the timing of gene expression for simultaneous or staggered studies. The enhanced flexibility opens new avenues for more complex and sophisticated studies, expanding the toolkit available for *T. brucei* functional genomics. Additionally, the possibility of introducing a third inducer element could further increase experimental flexibility, enabling more elegant and mechanistic studies, all within a single cell line.

## Supporting information

**S1 Table.  Primers used in this study.** Underline letter; restriction enzyme site.
(PDF)

**S2 Table.  Cell lines used in this study.** Superscript refers to the inducer used for the cell line. Tet, tetracycline; Van, vanillic acid; VaT, vanillic acid and tetracycline.
(PDF)

**S1 Fig.  Inductions with varying tetracycline concentrations.** (A) Growth curve of IBOE$^{Tet}$ grown in the absence and presence of 4 µg/ml Tet. Error bars represent ± s.d. of the mean from three biological replicates. Some error bars are too small to be displayed. (B) Western blot detection of IBWT and EF1α1 during a 10 day induction. $2 \times 10^6$ cell equivalents were loaded per lane. C, POLIB-PTP single allele tagged cell line. (C) Growth curve of IBComp$^{Tet}$ grown in the absence and presence of 1, 2 and 4 µg/ml Tet. Error bars represent ±s.d. of the mean from three biological replicates. Some error bars are too small to be displayed. (D) Western blot detection of PTP tag and EF1α1. $2 \times 10^6$ cell equivalents loaded per lane. C, POLIB-PTP single allele tagged cell line; $4 \times 10^6$ cell equivalents. (E) Northern blot of total RNA from IBComp$^{Tet}$. U, uninduced; 4, induced with 4 µg/ml Tet for 48 hr. Following probing for *TbPOLIB* (4.2 kb), the same blot was stripped and reprobed for *α-tubulin*.
(TIF)

**S2 Fig.  Expression of tetracycline and vanillic acid repressor in SMUMA.** (A) Western blot detection of Tet repressor and EF1α1 in 29-13 and SMUMA clonal cell lines. $2 \times 10^6$ cell equivalents loaded per lane. (B) Northern blot of total RNA from SMUMA clonal cell lines. Top, probing for the Van repressor; Bottom, EtBr-stained rRNAs as loading control. (C) Growth of SMUMA P2B7 clonal cell line in the absence and presence of 250 µM Van, 4 µg/ml Tet or a combination of both. Error bars represent ±s.d. of the mean from three biological replicates. Some error bars are too small to be displayed.
(TIF)

**S3 Fig. Specific and independent vanillic acid induction of *POLIB* RNAi.** (A) Northern blot of total RNA from IBComp<sup>VaT</sup> clonal line P5D1. U, uninduced; 2, induced with 250 µM Van for 48 hr. Following probing for *TbPOLIB* (4.2 kb), the same blot was stripped and reprobed for *POLIC, POLID.* EtBr-stained rRNAs as loading control. (B) Representative western blot detection of IBPTP and loading control EF1α1 from IBRNAi<sup>Van</sup>. C, single allele cell line endogenously expressing POLIB-PTP; U, uninduced; 2, induced with 250 µM Van for 48 hr. 5 x 10⁶ cell equivalents were loaded per lane.
(TIF)

**S4 Fig. Semi-automated kDNA volume analysis.** Representative images of asynchronous SMUMA cells. DNA was stained with DAPI (blue) and basal bodies were detected using YL1/2 (green). Multiple karyotypes are indicated in DAPI image. Scale bar, 20 µm. Quantification of kDNA volume from 580 DAPI stained cells. The red line indicates the mean of the kDNA volume for each karyotype. ****: *P* value <0.0001, as calculated by an unpaired two-tailed *t* test.
(TIF)

**S5 Fig. Southern blot analyses of free minicircles.** (A) Representative Southern blot showing the changes in free minicircles at selected induction points for IBComp<sup>VaT</sup> (clone P2D11) grown in the absence or presence of 250 µM Van. (B) Same as (A) but for IBComp<sup>VaT</sup> (clone P2D11) grown in the absence or presence of 250 µM Van and 4 µg/ml Tet. (C) Same as (A) but for IBComp<sup>VaT</sup> (clone P5D1) grown in the absence or presence of 4 µg/ml Tet. Abbreviations for all blots: k, kDNA network; N/G, nicked/gapped; MG, multiply gapped; CC, covalently closed; L, linearized; gDNA, loading control.
(TIF)

**S1 Raw Images. Original images for all gels and blots.**
(PDF)

## Acknowledgments

The authors thank members of our research group for their valuable comments and feedback throughout the conduct of this study and preparation of the manuscript. The authors also thank Ms. Lindsey Foster for helping to screen clonal cell lines for this study. The microscopy data was gathered in the Light Microscopy Facility (RRID:SCR_021148) and Nikon Center of Excellence at the Institute for Applied Life Sciences, UMass Amherst.

## Author contributions

**Conceptualization:** Raveen Armstrong, Stephanie B. Delzell, Michele M. Klingbeil.

**Formal analysis:** Raveen Armstrong, Michele M. Klingbeil.

**Funding acquisition:** Michele M. Klingbeil.

**Investigation:** Raveen Armstrong, Matt J. Romprey, Henry M. Raughley.

**Methodology:** Raveen Armstrong, Stephanie B. Delzell, Matthew P. Frost, James Chambers, Grace G. Garman, David Anaguano.

**Project administration:** Michele M. Klingbeil.

**Resources:** Raveen Armstrong, Stephanie B. Delzell, David Anaguano.

**Supervision:** Michele M. Klingbeil.

**Validation:** Matt J. Romprey, Henry M. Raughley.

**Visualization:** Raveen Armstrong, Michele M. Klingbeil.

**Writing – original draft:** Raveen Armstrong, Michele M. Klingbeil.

**Writing – review & editing:** Raveen Armstrong, Stephanie B. Delzell, James Chambers, Michele M. Klingbeil.

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
