## [Decision Letter · Decision Letter 0]

25 Mar 2025

PONE-D-25-11956An independently tunable dual control system for RNAi complementation in Trypanosoma bruceiPLOS ONE

Dear Dr. Klingbeil,

Thank you for submitting your manuscript to PLOS ONE. After careful consideration, we feel that it has merit but does not fully meet PLOS ONE’s publication criteria as it currently stands. Therefore, we invite you to submit a revised version of the manuscript that addresses the points raised during the review process.

We look forward to receiving your revised manuscript.

Kind regards,

Vyacheslav Yurchenko, Ph.D.

Academic Editor

PLOS ONE

“The authors thank members of our research group for their valuable comments and feedback throughout the conduct of this study and preparation of the manuscript. The authors also thank Ms. Lindsey Foster for helping to screen clonal cell lines for this study. The microscopy data was gathered in the Light Microscopy Facility (RRID:SCR_021148) and Nikon Center of Excellence at the Institute for Applied Life Sciences, UMass Amherst with support from the Massachusetts Life Sciences Center. This work was financially supported by Bridge Funding from the University of Massachusetts College of Natural Sciences to M.M.K., a UMass Amherst Graduate School Pre-Dissertation Research Grant awarded to RA and the Donald P. Reed Legacy Fund. The funders had no role in study design, data collection and analysis, decision to publish, or preparation of the manuscript. The authors declare no competing financial interest.”

Additional Editor Comments:

The manuscript on an independently tunable dual control system for RNAi complementation in Trypanosoma brucei was reviewed by 2 independent referees and they both were very positive about this work. Please add a few final touches (as requested by one of the reviewers) and it'll get accepted without re-review.

Reviewers' comments:

Reviewer's Responses to Questions

**Comments to the Author**

1. Is the manuscript technically sound, and do the data support the conclusions?

Reviewer #1: Yes

Reviewer #2: Yes

2. Has the statistical analysis been performed appropriately and rigorously? 

Reviewer #1: Yes

Reviewer #2: Yes

3. Have the authors made all data underlying the findings in their manuscript fully available?

Reviewer #1: Yes

Reviewer #2: Yes

4. Is the manuscript presented in an intelligible fashion and written in standard English?

Reviewer #1: Yes

Reviewer #2: Yes

5. Review Comments to the Author

Reviewer #1: This is a very nice contribution from two different perspectives - it establishes a dual inducible and tunable system for functional studies of Trypanosoma brucei genes, a major upgrade of a tool box that is in place for a almost 25 years and thanks to which this parasitic protist was lifted among top model organisms. So this reviewer simply applauds this and hopes mane labs will start using it.

Next, the authors used the newly developed system for smart functional dissection of POLIB, an unusual mitochondrial polymerase that has most likely specialized functions due to its association with the complex kinetoplast DNA network. The insight they present is novel, interesting and provide further solid grounds for acceptance of this study. Finally, they introduce Kinetometric - a new program to exactly evaluate the size of the kinetoplast in T. brucei, certainly another valuable addition to the field.

I am missing at least some newer references to kinetoplast studies; the authors cite old papers which is fine, but a fresh look into the literature (f.e. Michieletto NAR 2025 and below) would refresh the intro and/or discussion (f.e. when listing the kinetoplast associated proteins (around line 530 in the discussion), the authors ignore a paper by Pyrih J. et al. (Cell Reports), which identified a range of such proteins, and or the contribution by Cadena et al Curr. Biol. 2024 that associate first protein with the nabelschnur.

Minor issues:

56 - unicellular protist does not sound good, as by most definitions protist = unicellular

63 and 74 - I suggest the authors replace "novel trypanosome briology" (used twice), as the combination of words is somewhat awkward.

102 - a reference seems to be missing here

496 and 497 - hours are missing

564 - Sunter and colleagues would be correct

587 - Bruhn 2010 - kind of forgotten here

Reviewer #2: One of the established approaches for studying the structure-function analysis of proteins in Trypanosoma brucei is RNAi complementation in the single inducer tetracycline-On system. However, it is well known in the field that this single inducer system has not been successful for studies of several proteins. Here the authors present the establishment of an alternative vanillic acid-tetracycline dual inducible system for procyclic form T. brucei. In addition, to track the earliest reduction in kinetoplast DNA (kDNA) size, they developed a semiautomated 3D image analysis tool to measure kDNA volume which will be helpful to study kDNA replication defects. Overall, this is an excellent manuscript and the dual inducer system is a very valuable addition to the available toolkit for T. brucei and will also allow temporal regulation to modulate the timing of gene expression.

6. PLOS authors have the option to publish the peer review history of their article (what does this mean? ). If published, this will include your full peer review and any attached files.

**Do you want your identity to be public for this peer review?** For information about this choice, including consent withdrawal, please see our Privacy Policy .

Reviewer #1: **Yes: ** Julius Lukes

Reviewer #2: No

---

## [Author Response · Author response to Decision Letter 0]

2 Apr 2025

Vyacheslav Yurchenko, Ph.D.

Rebuttal responses for PONE-D-25-11956

April 2, 2025

Dear Dr. Yurchenko,

Thank you for your letter and the referee's comments on our manuscript. We were happy to learn that both reviewers were very positive about our work, indicated that we report interesting and novel findings that will be of value to the wider community.

We thank the reviewer's for their time and overall supportive comments. We also greatly appreciate the suggestions of Reviewer 1 to provide better references and rewording in some areas of the manuscript.

We believe we have addressed all of their concerns in a satisfactory way as outlined below.

Our responses are highlighted in blue and modifications to the manuscript are highlighted with yellow with appropriate line references for the revised manuscript.

Responses to Academic Editor

Thank you for pointing out our manuscript did not adhere to PLOS ONE's style requirements. We have addressed the following: 1. font size for headings and subheading has been corrected; 2. each paragraph is now indented; 3. titles for headings and subheadings have been corrected to sentence case; and 4. in text referencing to figure and supplemental files has been corrected.

All changes have been addressed throughout the manuscript.

2. Thank you for stating the following in the Acknowledgments Section of your manuscript: Please remove any funding-related text from the manuscript and let us know how you would like to update your Funding Statement.

Thank you for pointing this out to us. We have removed all text related to support from the Acknowledgments section and provided our amended funding statement to be submitted on our behalf in the cover letter; new Acknowledgments lines 655-660.

3. PLOS ONE now requires that authors provide the original uncropped and unadjusted images underlying all blot or gel results reported in a submission’s figures or Supporting Information files.

In your cover letter, please note whether your blot/gel image data are in Supporting Information.

We have provided all original blot/gel data as a supplemental pdf file (S1_raw_images). We mention this in our cover letter.

4. Please review your reference list to ensure that it is complete and correct. If you have cited papers that have been retracted, please include the rationale for doing so in the manuscript text, or remove these references and replace them with relevant current references.

We have carefully reviewed all references from our original submission and those that we have added for the revised version. None of the references have been retracted.

Responses to Reviewer # 1: (reviewers comments are italicized)

1. I am missing at least some newer references to kinetoplast studies; the authors cite old papers which is fine, but a fresh look into the literature (f.e. Michieletto NAR 2025 and below) would refresh the intro and/or discussion (f.e. when listing the kinetoplast associated proteins (around line 530 in the discussion), the authors ignore a paper by Pyrih J. et al. (Cell Reports), which identified a range of such proteins, and or the contribution by Cadena et al Curr. Biol. 2024 that associate first protein with the nabelschnur.

1a. We have added the Michieletto NAR 2025 reference to line 79 when referring to the overall structure of the kDNA.

1b.We have added the following text to highlight the unusual nabelschnur structure and 2 essential proteins for kDNA segregation (line 90-94) with the appropriate references;

Another distinctive feature is the final physical connection between daughter kDNA networks at the later stages of segregation called the nabelschnur (umbilical cord). This filamentous bridge contains maxicircles threads (Gluenz et al Mol Cell Biol 2011) with at least two essential proteins for kDNA segregation, a leucine aminopeptidase (LAP1) and NAB70 (Pena-Diaz et al PLos Path 2017; Cadena et al Curr Biol 2024).

1c. We have added two references (Billington et al Nature Microbio 2023; Pyrih et al Cell Reports 2023) to line 532 when referring to localization of proteins to multiple subcompartments around the kDNA network.

2. 56 - unicellular protist does not sound good, as by most definitions protist = unicellular

We agree and have removed the redundant word “unicellular”; line 57.

3. 63 and 74 - I suggest the authors replace "novel trypanosome briology" (used twice), as the combination of words is somewhat awkward.

We agree and replaced novel trypanosome with “eukaryotic”; line 63.

Additionally, we replaced novel trypanosome biology with “divergent trypanosome features”; line 75.

4. 102 - a reference seems to be missing here

Yes, we inadvertently omitted the following reference (Delzell et al Biochemistry 2022); line 107.

5. 496 and 497 - hours are missing

We have not made any changes here because we are referring to the total number of doublings and not the doubling time; lines 499 and 500.

6. 564 - Sunter and colleagues would be correct.

Thank you for suggesting this fix. However, when we carefully read through the Sunter MBP 2016 paper, pJ1173 was generated in this paper in which Jack Sunter was the sole author. Therefore, we will keep the original language of “original approach reported by Sunter”; line 569.

7. 587 - Bruhn 2010 - kind of forgotten here

Again thank you for catching this incorrectly formatted reference. This has now been updated appropriately; line 592.

Responses to Reviewer #2:

Thank you for your very positive comments!

Again thank for your time and very positive comments on our manuscript for publication in PLoS One.

---

## [Editor Report · Decision Letter 1]

4 Apr 2025

An independently tunable dual control system for RNAi complementation in Trypanosoma brucei

PONE-D-25-11956R1

Dear Dr. Klingbeil,

We’re pleased to inform you that your manuscript has been judged scientifically suitable for publication and will be formally accepted for publication once it meets all outstanding technical requirements.

Kind regards,

Vyacheslav Yurchenko, Ph.D.

Academic Editor

PLOS ONE

Additional Editor Comments (optional):

In my opinion, this is a very interesting and important work